# Controlling bimerons as skyrmion analogues by ferroelectric polarization in 2D van der Waals multiferroic heterostructures

Wei Sun [1], Wenxuan Wang[1], Hang Li [1✉], Guangbiao Zhang[1], Dong Chen[1], Jianli Wang[1] & Zhenxiang Cheng [2✉]

Atom-thick van der Waals heterostructures with nontrivial physical properties tunable via the magnetoelectric coupling effect are highly desirable for the future advance of multiferroic devices. In this work on LaCl/In$_2$Se$_3$ heterostructure consisting of a 2D ferromagnetic layer and a 2D ferroelectric layer, reversible switch of the easy axis and the Curie temperature of the magnetic LaCl layer has been enabled by switching of ferroelectric polarization in In$_2$Se$_3$. More importantly, magnetic skyrmions in the bimerons form have been discovered in the LaCl/In$_2$Se$_3$ heterostructure and can be driven by an electric current. The creation and annihilation of bimerons in LaCl magnetic nanodisks were achieved by polarization switching. It thus proves to be a feasible approach to achieve purely electric control of skyrmions in 2D van der Waals heterostructures. Such nonvolatile and tunable magnetic skyrmions are promising candidates for information carriers in future data storage and logic devices operated under small electrical currents.

[1] Institute for Computational Materials Science, School of Physics and Electronics, Henan University, 475004 Kaifeng, People's Republic of China. [2] Institute for Superconducting & Electronic Materials, Australian Institute of Innovative Materials, University of Wollongong, Innovation Campus, Squires Way, North Wollongong, NSW 2500, Australia. ✉email: hang.li@vip.henu.edu.cn; cheng@uow.edu.au

A magnetic skyrmion, usually a few nanometers in diameter, is a topologically protected magnetic quasi-particle with a whirling spin texture in real space[1]. Stable magnetic skyrmions, emerging in various magnetic materials[1–6] as a result of competing Heisenberg exchange and Dzyaloshinskii–Moriya interaction (DMI)[7], can be driven by low current density[8–13]. The creation and annihilation of magnetic skyrmions in thin films have been demonstrated by current[14,15], marking a significant step toward low-dimensional skyrmion systems. In recent experiments skyrmion and its motion under electric current have been observed in Fe₃GeTe₂[16,17], proving 2D magnetic materials a new category of skyrmion medium. In addition, another study showed that ferroelectric (FE) polarization can be integrated into the magnetic skyrmion systems to realize nonvolatile control via the magnetoelectric coupling effect in BaTiO₃/SrRuO₃ perovskite heterostructure (HS)[18]. It suggests that skyrmions can be controlled directly by electric fields—instead of electric current—via polarization switching. This significantly reduces energy consumption. However, whether this type of control can be extended to atom-thick van der Waals (vdW) HS remains largely elusive. Once achieved, it delivers great application potential for high-performance spintronic devices based on 2D skyrmions.

Since the discovery of graphene, extensive research on two-dimensional ferromagnetic (FM)[19–25] and FE[26–32] materials has been carried out rapidly. The magnetic anisotropy in 2D materials induced by spin–orbit coupling enables the long-range magnetic ordering, forming easy-axis or easy-plane 2D magnets. In the easy-axis 2D magnets, skyrmions are usually of the Néel-type, whereas in the easy plane 2D magnet, skyrmions usually exist in bimerons form[33–35]. When 2D FE and FM materials are combined to form atomic layer-thick multiferrous vdW HSs, such 2D system endows nonvolatile coupling between two ferroic orderings[36–38], compensating the scarcity of the single-phase multiferroic material. Furthermore, nonvolatile FE polarization controlled magnetic skyrmions through the magnetoelectric coupling effect is highly anticipated in such systems. In contrast to the traditional perovskite-based multiferroic HSs, 2D multiferroic HSs have an inherent advantage in achieving strong magnetoelectric coupling, that is, all atoms are exposed to the surface, which makes magnetism more sensitive to FE polarization. Recent reports have demonstrated that the interlayer magnetoelectric coupling in a 2D vdW HSs survives the large space between two different ferroic layers. For example, polarization can manipulate a variety of properties including the conductivity of the CrI₃ in the CrI₃/Sc₂Co₂ HS[36], the magnetic anisotropy of the CrGeTe₃ in CrGeTe₃/In₂Se₃ HS[37], and magnetic ordering of the FeI₂ in the FeI₂/In₂Se₃ HS[38]. All these works indicate the possibility of electric control of the magnetism and even skyrmions in 2D vdW HSs.

Although the existence of magnetoelectric coupling effect has been confirmed in the 2D vdW HSs, the influence of the polarization on the magnetism still lacks systematic investigation. Besides, the usual low Curie temperature of the 2D magnetic monolayer limits the study in terms of practical applications. In this work, we propose a LaCl/In₂Se₃ multiferroic HS, where In₂Se₃ is an ideal 2D FE material with controllable out-of-plane spontaneous polarization[24,39–41], while LaCl was synthesized decades ago and proved to be a vdW layer compound[42–45]. Furthermore, LaCl monolayer as an easy-plane FM metal has been confirmed in many works[46–49]. In addition, La-5d orbital has a strong spin–orbit coupling, which satisfies the requirement of producing strong Dzyaloshinskii–Moriya interaction, i.e., a necessary condition for the generation of magnetic chiral skyrmions. The nearly perfect lattice matching (lattice mismatching rate <0.1%) between the two vdW compounds makes them ideal model systems to construct HS for theoretical studies.

Our results show that the magnetic skyrmions in bimerons form is generated in the atom-thick vdW LaCl/In₂Se₃ multi-ferrous HS, see Fig. 1a, due to the broken inversion symmetry of LaCl by the In₂Se₃ FE polarization. The bimerons can be driven by current and generated or annihilated by FE switching. The diameter of bimeron is only about 23 nm, which enhances the controllability and integrability of the bimerons-based functional devices. In addition, we have systematically investigated the effect of the polarization orientation and magnitude on magnetism and realized the tuning of the Curie temperature and the magnetic easy axis. The strong magnetoelectric coupling effect observed in the 2D HS is unprecedented. Our results pave a new avenue for future devices based on vdW structures.

## Results

**Material model and computational details.** We combined first-principles density functional theory (DFT) and micromagnetic simulation to investigate the magnetism of LaCl monolayer and LaCl/In₂Se₃. Monte Carlo simulation was performed to locate the FM Curie temperature. The electronic properties and basic magnetic parameters of the LaCl monolayer and the LaCl/In₂Se₃ were calculated using the VASP package[50,51] based on the projected augmented wave (PAW) pseudopotentials. We constructed the HS along the [001] direction, see Fig. 1b, c, fully releasing the xy-plane lattice constant and spatial ion coordinates. The lattice constants of the free-standing LaCl and In₂Se₃ monolayer are, respectively, 4.033 Å and 4.035 Å—only 0.1% lattice mismatch rate, and we can thus safely neglect the influence of strain. The Heisenberg type spin Hamiltonian of LaCl can be expressed as,

$$H = -J_1 \sum_{i,j} \mathbf{M}_i \cdot \mathbf{M}_j - J_2 \sum_{i,k} \mathbf{M}_i \cdot \mathbf{M}_k - J_3 \sum_{j,l} \mathbf{M}_j \cdot \mathbf{M}_l - \mathbf{D}_1 \sum_{i,j} \mathbf{M}_i \times \mathbf{M}_j$$
$$- \mathbf{D}_2 \sum_{i,k} \mathbf{M}_i \times \mathbf{M}_k - \mathbf{D}_3 \sum_{j,l} \mathbf{M}_j \times \mathbf{M}_l - K \sum_i \left(M_i^z\right)^2,$$

$$(1)$$

where $J_1$, $J_2$, and $J_3$ are the exchange coupling parameters; and similar notation applies to the DM vectors ($\mathbf{D}_1$, $\mathbf{D}_2$, and $\mathbf{D}_3$), as shown in Fig. 1d. $\mathbf{M}$ denotes the magnetic moment of each atom, and $K$ is the perpendicular magnetic anisotropy constant. The calculation method of the exchange coupling parameter,

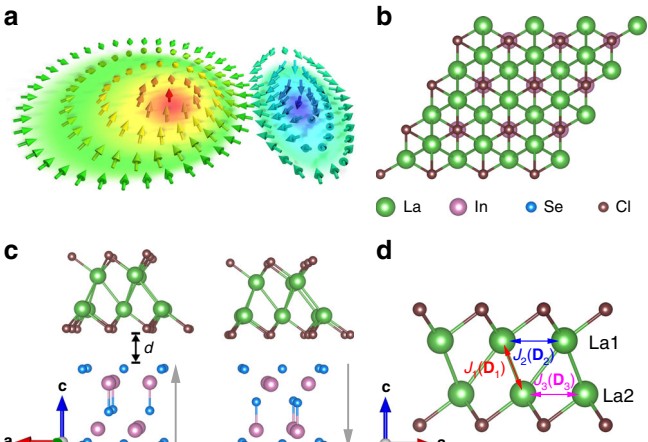

**Fig. 1 Crystal structure diagram of LaCl/In₂Se₃ HS. a** Illustrations of a magnetic bimeron. **b**, **c** show the top and side views of the HS, respectively. The gray arrow represents the polarization direction of In₂Se₃. **d** Magnetic structure of LaCl for calculating the exchange coupling parameter (DMI vector).

perpendicular magnetic anisotropy constant, and DM vectors are summarized in the Supplementary Information part1 and part2. Subsequently, the parameters extracted from DFT calculation were used subsequently as the input to the micromagnetic simulation using OOMMF software[52]. The boundary effect was taken into account by imposing the interfacial DMI in micromagnetic simulations[53].

The skyrmion is defined in terms of the nonzero integer topological charge defined as[12,54],

$$Q = \frac{1}{4\pi} \int \mathbf{m} \cdot \left( \frac{\partial \mathbf{m}}{\partial x} \times \frac{\partial \mathbf{m}}{\partial y} \right) dx dy, \tag{2}$$

where $\mathbf{m}$ is the normalized magnetization. When $Q$ is quantized at the value $\pm 1$, we may refer this state to a skyrmion.

In addition, we performed Monte Carlo simulations with a $50 \times 50 \times 1$ supercell based on the Heisenberg model to find the FM Curie temperature ($T_c$) of the LaCl monolayer and LaCl/In$_2$Se$_3$. For each temperature, $10^5$ Monte Carlo updates were employed. The FM Curie temperature $T_c$ is estimated by the peak position of the magnetic susceptibility (see Supplementary Information, Part 3).

**Micromagnetic simulation of bimeron.** Here we use $P+$ ($P-$) to represent the polarization direction of the In$_2$Se$_3$ that is along the $+z$ ($-z$) axis. The result shows that the free-standing LaCl monolayer is metallic with in-plane magnetization, consistent with previous reports[55–57]. In the HS, the interlayer spacing $d$ between LaCl and In$_2$Se$_3$ is 3.58 and 3.17 Å in LaCl/$P+$ and LaCl/$P-$, respectively, indicating the non-bonded nature. Moreover, both LaCl/$P+$ and LaCl/$P-$ structures remain in the metallic FM state, the same as that in the monolayer LaCl, showing that the switching of the In$_2$Se$_3$ polarization does not directly affect the magnetic ordering in LaCl. However, the exchange interaction parameters are more sensitive to the polarization. We summarize in Table 1 the calculated exchange interaction parameters. In the $P+$ state, $J_2$ and $J_3$ are enhanced compared to that of the free-standing LaCl monolayer, whereas the magnitude of $J_1$ sees no evident change. Upon polarization reversal to the $P-$ state, $J_1$ undergoes larger enhancement than $J_2$ and $J_3$. In addition, the FE polarization that originates from the In$_2$Se$_3$ layer breaks the spatial inversion symmetry in the adjacent LaCl, which gives rise to a DMI in HS[7,58]. We present in Table 1 the calculated in-plane $\mathbf{D}$ vector of the HS. The results show that the magnitude of $\mathbf{D}$ vector is different in two polarization states, indicating nonvolatile control of the exchange effect and DMI by the FE polarization. The DMI induced in our 2D HS by ferroelectric polarization is superior to the DMI induced in the CrI$_3$ monolayer by an electric field that breaks its spatial inversion symmetry[59]: In In$_2$Se$_3$ the nonvolatile nature of FE polarization persists into the magnetic switching in the LaCl/$P\pm$ structure with a significant reduction of energy consumption in the proposed device.

We used OOMMF software to perform micromagnetic simulations of the LaCl monolayer and LaCl/$P\pm$ HSs in a nanodisk geometry with a diameter of 200 nm. The varied parameters including $J$, $\mathbf{D}$, $K$, and $M_s$ for $P+$ and $P-$ states are listed in Table 1, respectively. The initial spin state was set to paramagnetic (random) states. In the LaCl/$P+$ configuration, bimerons spin texture emerges, as shown in Fig. 2a. The calculation shows the topological charge of the bimeron $Q = 1.0$, consisting of a vortex ($Q = 0.5$) and an anti-vortex ($Q = 0.5$). The extracted bimeron texture map is depicted in Fig. 2d, e. Due to the presence of in-plane anisotropy, bimeron's outer magnetization is along the in-plane direction rather than the out-of-plane one, which consequently allows two bimerons with opposite topological numbers ($Q = 1$ and $Q = -1$) to coexist in the same magnetic domain (see Supplementary Fig. 4)[60]. This is significantly different from other types of skyrmions. This spin texture disappears, accompanied by a vanishing $Q$ to zero when the polarization is reversed to the $P-$ state, shown in Fig. 2b. Subsequently, we found that bimeron can appear when the $K$ value ranges from $-0.04$ to $-0.13$ meV (other parameters remain unchanged). Figure 2c represents the bimeron texture when the $K = -0.1$ meV, showing a smaller bimeron size compared to the $P+$ state and this phenomenon is explained in part 5 of the Supplementary Information. In the $P-$ state, the enhanced $K$ value caused by the polarization completely exceeds the range that allows bimeron to exist, leading to bimeron's annihilation in this polarized state. Furthermore, Fig. 2c verifies the above conclusion that bimerons can appear in the same magnetic domain. Therefore, by simply switching the FE polarization, two distinct magnetic states—referred to in the binary code as "0" and "1"—can be realized and could be used as building blocks for information storage. The diameter of bimeron in LaCl/$P+$ HS is only about 23 nm, which enhances the controllability and integratability of the bimerons-based functional devices.

We show in Fig. 3 the magnetization as a function of $K$ while keeping other parameters the same as those in the LaCl/$P+$ configuration. When $K$ is positive, the perpendicular magnetic anisotropy provides an easy axis to support magnetizations pointing either up or down. In this case, the formation of Néel-type skyrmions is energetically favored. When $K$ is negative, however, the easy axis is confined within plane by the in-plane anisotropy, forcing the Néel-type skyrmions to evolve into vortices, the FM stripes into anti-vortices, and eventually the Néel-type skyrmions transform into bimerons.

The current-driven motion of bimeron in the LaCl/$P+$ configuration was simulated on a $150 \times 600$ nm nanotrack. Prior to the injection of current, we created and relaxed the bimeron near the left end of the nanotrack. An in-plane current of $j = 3 \times 10^{10}$ A m$^{-2}$ was later applied along the $x$ direction to mobilize the bimeron, see Fig. 2g. The bimeron remains stable while traversing the nanotrack at the speed of 72.17 m s$^{-1}$; the transverse motion along the $y$-axis is due to the skyrmion Hall effect[11,61].

**Regulation of LaCl magnetism by In$_2$Se$_3$ polarization.** In our system, the out-of-plane polarization of the In$_2$Se$_3$ layer redistributes charges in the LaCl layer to screen the polarization field, resulting in a change in the magnetic properties of LaCl. The enhanced polarization is thus expected to improve the regulation for LaCl magnetism. For the ferroelectric substrate In$_2$Se$_3$, polarization increases as a function of the film thickness and saturates as the thickness increases to three layers[26]. Therefore, using a three-layered In$_2$Se$_3$, we simulated an enhanced polarization which is denoted as $3P\pm$ for simplicity. Note that the method by increasing the layer thickness for better effect does not apply to all 2D HS system. For example, in the WTe$_2$/CrI$_3$ HS, the conductance of WTe$_2$ depends on the nearest CrI$_3$ layer and is not affected by the CrI$_3$ thickness[62]. The impact of the polarization intensity in In$_2$Se$_3$ on the magnetism in LaCl is shown in

**Table 1 Magnetic parameters of LaCl/In$_2$Se$_3$ HS.**

|  | $J_1$ | $J_2$ | $J_3$ | $D_1$ | $D_2$ | $D_3$ | $K$ | $M_s$ |
|---|---|---|---|---|---|---|---|---|
| LaCl/$P+$ | 1.36 | 8.32 | 8.28 | 0.58 | 0.78 | 0.84 | −0.19 | 0.88 |
| LaCl/$P-$ | 3.30 | 5.93 | 4.92 | 0.23 | 0.42 | 0.31 | −0.76 | 0.93 |

Summary of exchange coupling parameter ($J$, in meV), in-plane DM vectors (D, in meV), perpendicular magnetic anisotropy constant ($K$, in meV), and saturation magnetic moment ($M_s$, in μB).

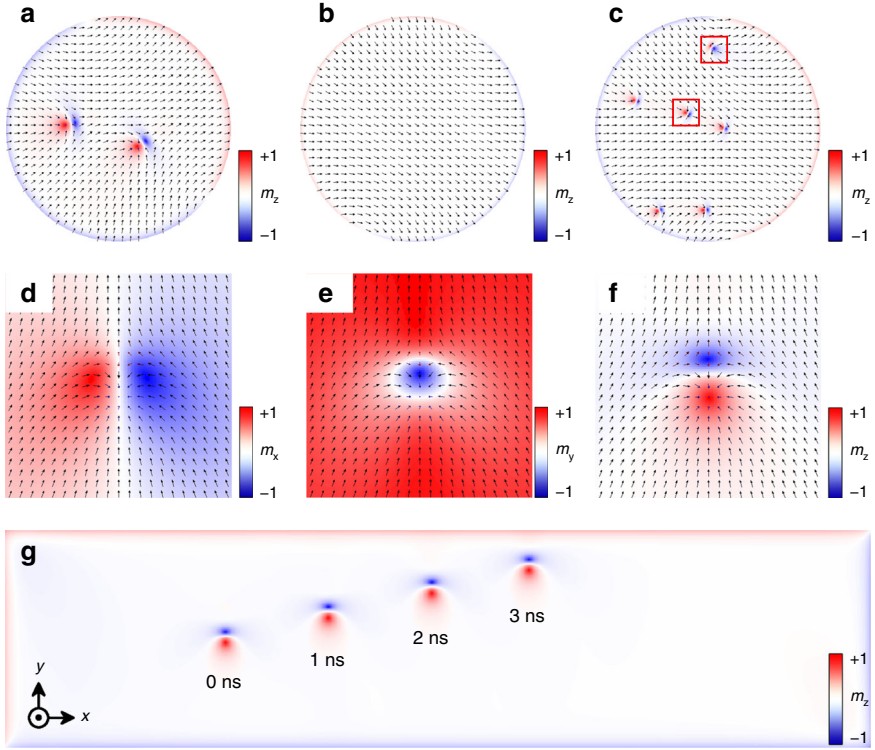

**Fig. 2 Ferroelectric regulation and current drive of bimeron.** The top views of the micromagnetic simulation for (**a**) LaCl/$P+$ and (**b**) LaCl/$P-$ HSs. **c** The top views of the micromagnetic simulation for LaCl/$P-$ HS with $K = -0.1$ meV, the red rectangle shows two bimerons with opposite topological numbers. **d–f** Blue–red color represents the region of $x$, $y$, $z$ component along the magnetic direction of a bimeron. **g** Snapshots of the bimeron moving under a current of $j = 3 \times 10^{10}$ A m$^{-2}$ on nanotracks.

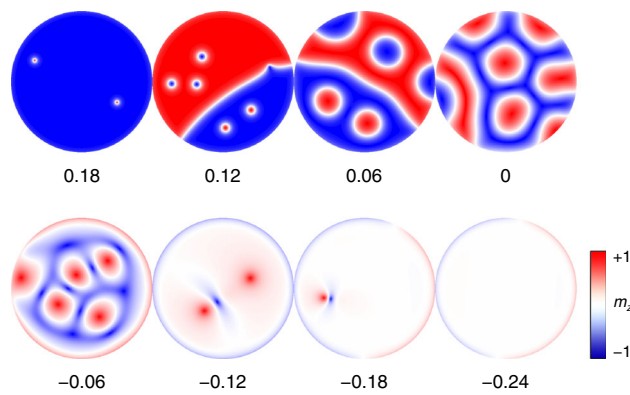

**Fig. 3 Evolutions of spin textures under the modulations of magnetic anisotropy.** Relaxed magnetic configurations under various $K$-values.

Fig. 4a, b. We found that $J_1$ and $K$ increase significantly with the polarization intensity along the $-z$ direction; both reach its maximum at $3P-$, 348 and 449% increase with respect to that of LaCl monolayer, respectively. When the polarization switches to the $+z$ direction, the in-plane exchange interaction parameters $J_2$ and $J_3$ have enhanced in the LaCl/$3P+$ configuration compared to the monolayer LaCl. $K$ decreases gradually with a decreasing polarization, and the easy-axis switches to out-of-plane in the LaCl/$3P+$ configuration. Therefore, the polarization vector offers a channel via which we may tune the exchange coupling and magnetic anisotropy. As a result of such correlation, the Curie temperature of LaCl shifts with the polarization switching, which is confirmed by the Monte Carlo simulation shown in Fig. 4c. We are thus able to tune the magnetism of the HS from FM LaCl/($3)P+$ to paramagnetic LaCl/($3)P-$ by the polarization when the temperature is in the range between 92 K (102 K) and 99 K

(110 K). They discuss the $T_c$ in detail in part 3 of the Supplementary Information. The magnetoelectric coupling in our system is robust and the two FE polarization states are stable: One can be switched into the other by an external electric field, yet either can survive the field removal. This bistability of FE polarization of In$_2$Se$_3$ means that the magnetic properties of the LaCl layer can be switched steadily between the two states. This feature is highly desirable in nonvolatile information storage.

Moreover, we found that the net magnetic moment is highly correlated with the charge transfer between LaCl and In$_2$Se$_3$, which is shown in Fig. 4d, e. The decrease in the electron number on the LaCl side is beneficial to the increase in the net magnetic moment of the system. In the LaCl/($3)P-$ configuration in particular, compared to LaCl/($3)P+$, more electrons participate in charge transfer to In$_2$Se$_3$, resulting in a significant increase in the net magnetic moment of the system. As shown in Fig. 5a, the spin-up states in the LaCl/$3P-$ configuration is populated more than the spin-down state. Viewing the decrease in the electron number on the LaCl side, we are able to conclude that the charge transfer across the interface from LaCl to In$_2$Se$_3$ occurs mainly in the spin-down state.

**The mechanism for interfacial multiferroicity.** Change in electronic structure holds the key to understand the variation in magnetism induced by polarization switching. As the differential charge density distribution in Fig. 5c shows, the tuning of the magnetism can be interpreted by electronic reconfiguration that takes place on the LaCl side. In the LaCl/$3P+$ configuration, pushed "down" by the FE polarization along $+z$, the electrons on the LaCl side are transferred from the top to the bottom surface of the entire HS, shown in Fig. 5b. While in the case of LaCl/$3P-$, the opposite polarization brings "up" electrons in LaCl from the lower surface. However, unlike the case of LaCl/$3P+$, electrons

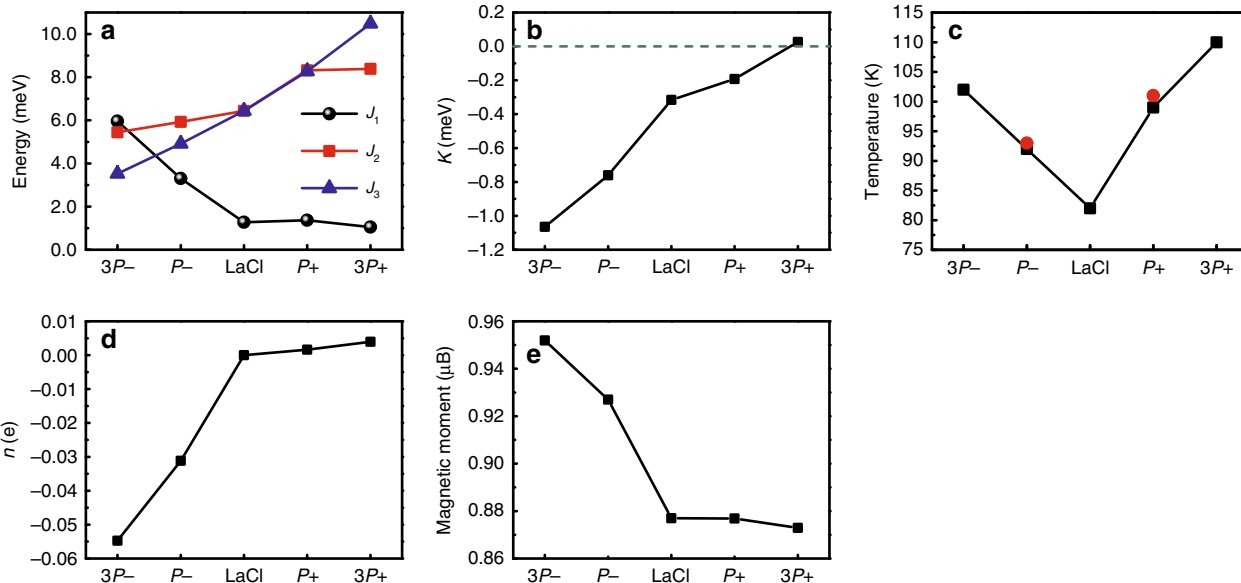

**Fig. 4 The influence of polarization intensity on magnetic parameters. a** Exchange interaction parameters in LaCl. **b** Perpendicular magnetic anisotropy constant in LaCl (**c**) Curie temperature ($T_c$) of LaCl. Since we only calculated the DMI of LaCl/$P\pm$ , the effects of the perpendicular magnetic anisotropy and exchange interaction terms on $T_c$ are presented by the data in black squares, and the effect of DMI on $T_c$ in LaCl/$P\pm$ is presented by the data in red spheres. **d** The number of electrons gained of LaCl. **e** The net magnetic moment of LaCl. In these figures, $3P\pm$ represents a configuration that contains three $In_2Se_3$ layers.

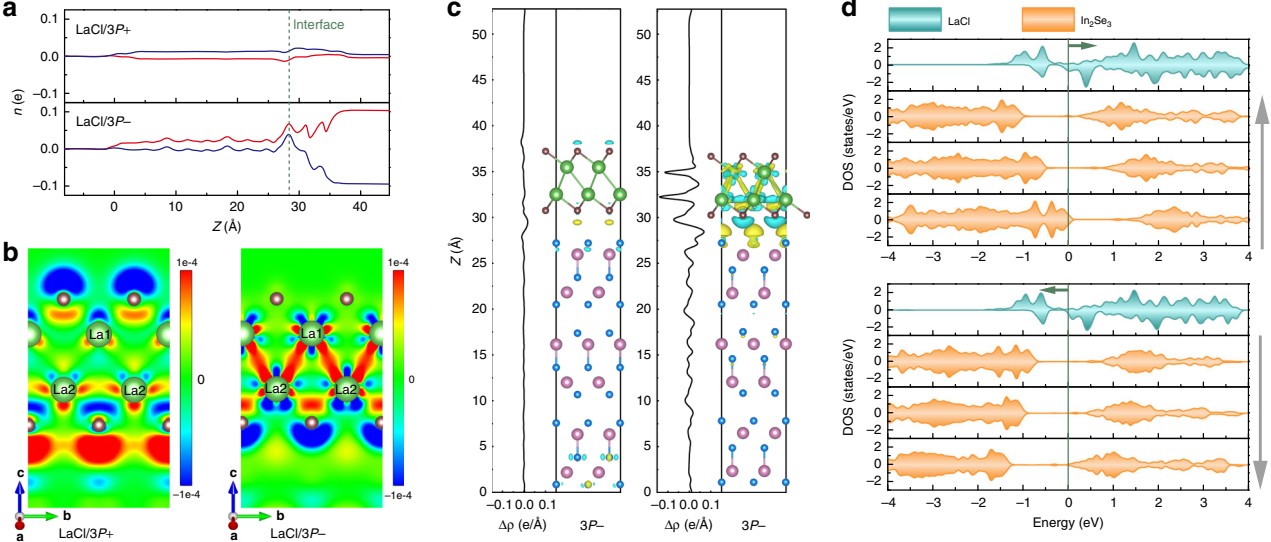

**Fig. 5 Charge transfer in different polarization states. a** Spin-up (red line) and -down (blue line) contributions on the charge transfers in LaCl/$3P\pm$ HSs. **b** Differential charge density on the LaCl side. **c** Differential charge density distributions of LaCl/$3P\pm$ HSs and integrals of differential charge densities (black line). **d** The layer-resolved partial DOS of LaCl/$3P\pm$ HSs. The gray and green arrows represent the polarization direction of $In_2Se_3$ and the moving direction of Fermi level in HSs, respectively.

are not directly transferred to the upper surface. Instead, it is concentrated in between La1 and La2. Different charge distribution ultimately affects the exchange coupling constant $J$ on the LaCl side, rendering it different in the $3P+$ and $3P-$ states. As per polarization switching, the electrons around the La ion are redistributed, leading to variation in the spin–orbit coupling of the La atom, too. This eventually gives rise to the variation of the DMI on the LaCl side.

We performed a layer-resolved partial density calculation on the LaCl/$3P\pm$ configurations to scrutinize the underlying mechanism. Figure 5d shows that the electronic distribution on the LaCl side, driven by the built-in electric field of $In_2Se_3$,

remains largely unchanged yet the Fermi energy has been shifted, which is due to the change of electric potential caused by the polarization discontinuity. In addition, owing to the broken time-reversal symmetry in ferromagnetic (LaCl part) materials, the charge transfer between LaCl and $In_2Se_3$ is different for spin-up and spin-down states, which results in magnetic states change in the two states that are polarized oppositely. The above analysis reveals that the change in ferromagnetism in LaCl is mainly driven by two mechanisms: (i) change in potentials caused by the polarization discontinuity, and (ii) time-reversal symmetry breaking in magnetic materials leads to a difference in charge transfer for spin-up and spin-down states. These two

requirements for polarization tuning of magnetism in HS are not limited to the present $LaCl/In_2Se_3$ system and we foresee that effective control of 2D ferromagnetism can be realized in other materials, too.

We must nevertheless point out that the surface layer of $In_2Se_3$ in the LaCl/3 $P+$ configuration exhibits metallic behavior and it falls into the category of polar metal. The screening effect in metal eliminates the possibility of FE switching in bulk materials. This screening is, however, negligible in an atomically thin film due to the strong penetration of the external electric field. The switchable behavior of 2D polar metals has been demonstrated experimentally[29]. Switching the polarization vector using an external electric field is therefore a viable solution.

In summary, we designed and investigated a 2D vdW LaCl/$In_2Se_3$ HS. Magnetic skyrmions can exist in bimerons form therein with a diameter of about 23 nm. By switching the polarization in $In_2Se_3$, the anisotropy, FM Curie temperature, and the bimerons were manipulated. Such a robust magnetoelectric coupling effect that occurs in a 2D vdW HS is unprecedented. Further analysis shows that the coupling between FM and FE is attributed to the effect of the polarization discontinuity in the FE substrate and the broken time-reversal symmetry in the magnetic film. Therefore, such tunability in magnetic properties achieved in the $LaCl/In_2Se_3$ HS can be transplanted to other material systems, too. The significance of this work is twofold. It not only demonstrates the feasibility of nonvolatile control for magnetism and skyrmions in $LaCl/In_2Se_3$ HS but also puts forward a general idea to manipulate many characteristics of the 2D vdW HSs by two oppositely polarized states. Furthermore, this work highlights that an artificially designed 2D multiferroic HS is an ideal platform to realize large ME coupling and observe nontrivial physical properties.

## Methods

**The DFT method and parameters.** First-principles calculations were performed using the VASP package based on the projected augmented wave pseudopotentials[50,51]. The electronic exchange-correlation potential is treated within the spin-polarized generalized gradient approximation plus $U$ (GGA + $U$) of PBEsol formula[50,63–65]. Due to that, the GGA algorithm will underestimate the bandgap of the $f$ orbitals, an effective 7 eV Hubbard $U_{eff}$ parameter is applied on La's $f$ orbitals using the Dudarev method[66] in order to correct its bandgap and prevents $f$ orbitals from participating in orbital hybridization near the Fermi level. We take the van der Waals corrections as parameterized in the semiempirical DFT-D3 method into consideration for all the configurations[67]. The first Brillouin zone is sampled with the $9 \times 9 \times 1$ $k$-points meshes for the DFT calculation both of the monolayer and the HSs. In order to ensure the energy convergence, denser $k$-points meshes up to $15 \times 15 \times 1$ are also tested, which shows that $9 \times 9 \times 1$ settings are enough to achieve the desired accuracy. A plane-wave basis set with a kinetic energy cutoff of 500 eV is employed. All structures are fully relaxed until the energy and force converge to $10^{-5}$ eV and $10^{-2}$ eV/Å, respectively. To eliminate the periodic boundary effect, a 15 Å thinness vacuum layer is introduced along the $z$ direction. Using spin–orbit coupling for calculating magnetic anisotropic energy and DMI.

**OOMMF simulations parameters.** OOMMF software were used to simulate the magnetization dynamics[52]. The time evolution of the magnetization is described by the Landau–Lifshitz–Gilbert (LLG) equation[68].

$$\frac{d\mathbf{M}}{dt} = -\gamma_0 \mathbf{M} \times \mathbf{h}_{eff} + \alpha \left( \mathbf{M} \times \frac{d\mathbf{M}}{dt} \right), \quad (3)$$

where $\mathbf{h}_{eff} = -\delta H/\delta \mathbf{M}$ is the effective field, $\alpha$ is the Gilbert damping coefficient, and $\gamma_0$ is the absolute gyromagnetic ratio. We include in the simulation a Slonczewski-like spin-transfer torque (STT) driven by the spin current that is generated by the spin-Hall effect,

$$\tau = -\frac{\gamma_0 \hbar j P}{2ae\mu M_S} \mathbf{M} \times (\mathbf{M} \times \mathbf{p}). \quad (4)$$

Parameter $e$ is the electron charge and $M_S$ is the saturation magnetic moment. $j$ is the electric current density. Spin-Hall angle $P = 0.4$. The lattice constant $a = 4$ Å. The polarization of spin current is $\mathbf{p} = -x$. The parameters of LaCl monolayer and $LaCl/In_2Se_3$ HS are shown in Table 1. In simulations models, we keep $\alpha = 0.2$ and $\gamma_0 = 2.211 \times 10^5$ m $A^{-1}$ $s^{-1}$. The size of the cells is chosen to be 4 Å × 4 Å × 5 Å.

## Data availability

The data that support the findings of this study are available from the corresponding authors upon reasonable request.

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

## Acknowledgements

We acknowledge grants from the National Natural Science Foundation of China under research (Nos. 51571083 and 11674083) and the Foundation of Postgraduate Education Innovation and Quality Improvement Project of Henan University (No. CX3040A0920215). Z.X.C. thanks Australia Research Council for support (DP190100150). H.L. acknowledges the support from the National Natural Science Foundation of China (Grant No. 11804078) and Henan University (Grant No. CJ3050A0240050).

## Author contributions

Z.X.C. conceived the idea. W.S., Z.X.C., and H.L. carried out the calculation and analysis of the result. W.S., Z.X.C., and H.L. wrote the paper. All authors contribute the comments on the paper.

## Competing interests

The authors declare no competing interest.
