## [Peer Review File · Nature Communications]

Reviewers' Comments:

Reviewer #1:

Remarks to the Author:

Authors studied a multiferroic vdW heterostructure LaCl/In₂Se₃ which can accommodate nonvolatile skyrmions and provide electrical tunability. The vdW multiferroicity has been a hot research interest and many relevant papers are being published. Nevertheless a current paper goes one step beyond what have been reported in most papers. Most have focused on the long range spin ordering, but this work studies a short range behavior like skyrmions. By using the exchange constants calculated by the DFT method, they showed emergence or disappearance of novel magnetic phases by ferroelectric switching. Thus I recommend the current work to be published. But minor revision is needed, as listed below.

1. Which atomic orbital of La gives a finite spin moment? In page 3 it says "La-4d orbital has a strong spin-orbit coupling", but it is confusing as the U_{eff} is applied to La f orbital.
2. In the DFT method, what does it mean by "The products of the lattice parameters and the k-points are set to be greater than 30 when sampling in the first Brillouin zone". Was the k-grid fine enough to ensure the convergence of all the calculated exchange constants?
3. I could not find the current-driven result. It is referring Figure 4 in page 7, but it seems not.
4. In Figure 4(c), the Curie temperature is enhanced for either type of HS. What will be origin? In their MC simulation to estimate the Curie temperature, it says 50x50x2 supercell. What is '2' along z-direction? Two atomic layers or LaCl bilayer?

Reviewer #2:

Remarks to the Author:

In this work, the authors theoretically demonstrated the control of skyrmions and magnetic properties in ferromagnetic LaCl layer by changing the ferroelectric polarization of In₂Se₃ layer in a multiferroic LaCl/In₂Se₃ heterostructure. The mechanism for multiferroicity was further investigated and the authors pointed out that the tuning of magnetism is caused by electronic reconfiguration in LaCl due to the polarization reversal in In₂Se₃. These can be interesting results but the reported data are not very convincing. Therefore at this point, I don not recommend its publication. Below are a few points I would like the authors to address.

1. The authors described J_1 as interlayer exchange constant and J_2 , J_3 as intralayer exchange constants. However, as shown in the figure 1d, all three exchange constants are associated with La atoms within the same LaCl layer. Normally the interlayer interaction represents the coupling between two individual layers in a van der Waals material. Can the authors comment on this?
2. In LaCl/P- configuration, skyrmions can be created or annihilated by tuning the anisotropy energy K from -0.5 meV to -0.76 meV. Since the appearance of skyrmion is very sensitive to the calculated parameters, it is hard to believe that the control of skyrmion using ferroelectric polarization can be as good as claimed by the authors.
3. Figures 2a and 2c show the simulation of skyrmions for LaCl/P+ and LaCl/P- with $K = -0.5$ meV. Why do the skyrmions for the two cases have different sizes?
4. In the ferroelectric polarization dependent LaCl magnetic properties section, the authors simulated an enhanced polarization of 3P using three layers of In₂Se₃. Since the multiferroic property combining FM and FE materials has a nature of proximity effect which is very sensitive on interlayer distance, is it justified to simply add up the ferroelectric polarization of three In₂Se₃

layers and consider it the polarization experienced by the LaCl layer. In a Zhao et al, Nature Materials, 19, 503–507(2020), the electric properties of WTe₂ layer is affected by tuning the magnetization of CrI₃ layer. From their results, the electric conductance of WTe₂ is majorly dependent on the nearest CrI₃ layer.

5. In figure 4 a, b and e, it is shown that the exchange coupling constants, the magnetic anisotropy energy and magnetic moment are largely affected when the polarization is applied in the -z direction and relatively insensitive when the polarization is aligned in the +z direction. In contrast, the Curie temperature increases symmetrically with increasing polarization magnitude in both directions. Since Curie temperature depends on the magnetic properties such as exchange energy, why the magnetic properties (exchange coupling, magnetic anisotropy and magnetic moment) and Curie temperature behave differently when ferroelectric polarization is applied in different directions?

6. There are many typos, grammatical errors and wrongly labelled figure indexes in the manuscript. For example, 1. on line 1, page 8, "to mobilize the bimeron, see Fig. 4." Fig. 4 should be Fig. 2g; 2. Figure caption of fig 2f should be 2g; 3. On last line, page 9, "underling" should be underlyling; etc. Please check the manuscript carefully and correct all the errors.

Reviewer #3:

Remarks to the Author:

The authors reported their theoretical findings that skyrmions can be generated, manipulated, and destroyed in heterostructures consisting of a ferromagnetic monolayer LaCl and a ferroic layer In₂Se₃. They combined 1) DFT calculations, 2) OOMMF micromagnetic simulations, and 3) finite temperature Monte Carlo simulations to show that, by controlling the polarization state of the heterostructure, one can manipulate T_c , the type of anisotropy, and skyrmion populations of the heterostructure. Interestingly, skyrmions are found to exist in the form of bimerons when LaCl has easy-plane anisotropy and they can be driven by a current. Creation/annihilation/manipulation of skyrmions in 2D magnets is an important topic in spintronics and using the heterostructures of 2D magnets and 2D ferroics is a promising direction, and I believe that this work reports the first comprehensive theoretical prediction of a suitable material for that purpose. Therefore, I recommend the publication of the manuscript in Nature Communications. However, I have a couple of minor comments, which should be addressed before the publication.

1) English usage is poor. Extensive language editing seems necessary.

2) Two important references are missing. [Park et al., "Néel-type skyrmions and their current-induced motion in van der Waals ferromagnet-based heterostructures," arXiv:1907.01425], where Néel type skyrmions are experimentally observed and driven in 2D magnet / h-BN structures, and [Ding et al., "Observation of Magnetic Skyrmion Bubbles in a van der Waals Ferromagnet Fe₃GeTe₂," Nano Lett. 20, 868 (2020)] where skyrmions are observed in a 2D ferromagnet.

3) About the current-induced skyrmion motion, it is not clear what type of spin torque is used. Is it spin-transfer torque or spin-orbit torque?

4) The discussion of the topological charge of observed skyrmions is missing. For example, a) can skyrmions of both $Q = 1$ and $Q = -1$ be created? Or only one type of skyrmions can be created? Also, b) What effects does changing the polarization have on the topological charge of stable skyrmions?

Dear Editor and Referees,

First of all, we thank Editor for his effort in delivering such prompt communication regarding our manuscript NCOMMS-20-20354A.

We are delighted by Referees' recommendation for publication. Meanwhile, we'd like to extend our gratitude to all Referees for their valuable comments and questions that are helpful in the revising the manuscript and repositioning our arguments.

We have pondered the questions and comments in the Referee's reports and our response to them is detailed in the second part of this Reply Letter and highlighted in red in the revised manuscript that is enclosed in the resubmission.

As Editor and Referee may soon find out, language in the revised manuscript has been largely improved. We also made significant effort to make sure the writing is suitable for a broad audience. To this end, we believe that the revised manuscript meets all the criteria for publication in Nature Communications.

Sincerely yours,
All the authors

Reply to Referees

Reply to the first Referee's report:

Authors studied a multiferroic vdW heterostructure LaCl/In₂Se₃ which can accommodate nonvolatile skyrmions and provide electrical tunability. The vdW multiferroicity has been a hot research interest and many relevant papers are being published. Nevertheless a current paper goes one step beyond what have been reported in most papers. Most have focused on the long range spin ordering, but this work studies a short range behavior like skyrmions. By using the exchange constants calculated by the DFT method, they showed emergence or disappearance of novel magnetic phases by ferroelectric switching. Thus I recommend the current work to be published. But minor revision is needed, as listed below.

Reply: We thank Referee for her/his recommendation for publication.

1. Which atomic orbital of La gives a finite spin moment? In page 3 it says “La-4d orbital has a strong spin-orbit coupling”, but it is confusing as the U_{eff} is applied to La f orbital.

Reply: Nonzero spin moment arises from partially filled orbitals. The La-5d orbit gives a finite spin moment since it is partially filled. Furthermore, GGA algorithm usually underestimates the band gap of the f orbitals, we adopted a relatively large Hubbard U_{eff} parameter of 7 eV on La- f orbital in order to correct its value of band gap, which prevents the f -orbitals from participating in orbital hybridization near the Fermi level. In addition, spin-orbit coupling is attributed to the La-5d orbital rather than La-4d orbital. Relevant discussion has been revised in our manuscript (Page 11, Paragraph 3, Line 4-7)”

“Due to that the GGA algorithm will underestimate the band gap of the f orbitals, an effective 7 eV Hubbard U_{eff} parameter is applied on the La’s f orbitals using the Dudarev method [68] in order to correct its band gap and prevents f -orbitals from participating in orbital hybridization near the Fermi level.”

2. In the DFT method, what does it mean by “The products of the lattice parameters and the k -points are set to be greater than 30 when sampling in the first Brillouin zone”. Was the k -grid fine enough to ensure the convergence of all the calculated exchange constants?

Reply: We thank Referee for bringing up this question. The first Brillouin zone is sampled with $9 \times 9 \times 1$ k -points meshes for the DFT calculation in both monolayer and the HSs. Denser k -points meshes up to $15 \times 15 \times 1$ are also tested to check the energy convergence. We found that the $9 \times 9 \times 1$ setting is sufficient to achieve the desired accuracy. To elucidate this point, we added in the revised manuscript the following explicit description (Page 12, Paragraph 1, Line 4-7):

“The first Brillouin zone is sampled with the $9 \times 9 \times 1$ k -points meshes for the DFT calculation both of the monolayer and the HSs. In order to ensure the energy convergence, denser k -points meshes up to $15 \times 15 \times 1$ are also tested, which shows that $9 \times 9 \times 1$ settings are enough to achieve the desirable accuracy.”

3. I could not find the current-driven result. It is referring Figure 4 in page 7, but it seems not.

Reply: We thank Referee for this comment. The current-driven result is shown in Fig. 2g. We added in the revised manuscript (Page 8, Paragraph 1, Line 2) the following

description and revised the captions accordingly.

“An in-plane current of $j = 3 \times 10^{10}$ A m⁻² was later applied along the x direction to mobilize the bimeron, see Fig. 2g.”

4. In Figure 4(c), the Curie temperature is enhanced for either type of HS. What will be origin? In their MC simulation to estimate the Curie temperature, it says 50x50x2 supercell. What is ‘2’ along z -direction? Two atomic layers or LaCl bilayer?

Reply: The number ‘2’ along the z -direction refers to two ‘atomic layers’ rather than ‘supercell’. To avoid ambiguity, the expression of this part has been carefully revised to $50 \times 50 \times 1$ supercell (Page 5, Paragraph 3, Line 1).

“we performed Monte Carlo simulations with a $50 \times 50 \times 1$ supercell based on the Heisenberg model to find the FM Curie temperature (T_c) of the LaCl monolayer and LaCl/In₂Se₃.”

Reply to the second Referee’s report:

In this work, the authors theoretically demonstrated the control of skyrmions and magnetic properties in ferromagnetic LaCl layer by changing the ferroelectric polarization of In₂Se₃ layer in a multiferroic LaCl/In₂Se₃ heterostructure. The mechanism for multiferroicity was further investigated and the authors pointed out that the tuning of magnetism is caused by electronic reconfiguration in LaCl due to the polarization reversal in In₂Se₃. These can be interesting results but the reported data are not very convincing. Therefore at this point, I don not recommend its publication. Below are a few points I would like the authors to address.

Reply: We truly thank the referee for taking out time to carefully read our manuscript and make constructive comments. The comments are indeed helpful to strength the arguments and improve accessibility and readability of manuscript. We have followed the referee’s suggestion and revised the previous manuscript.

1. The authors described $J1$ as interlayer exchange constant and $J2, J3$ as intralayer exchange constants. However, as shown in the figure 1d, all three exchange constants are associated with La atoms within the same LaCl layer. Normally the interlayer

interaction represents the coupling between two individual layers in a van der Waals material. Can the authors comment on this?

Reply: To clarify the definitions of different exchange coupling parameters, we have revised the relevant description (Page 5, Paragraph 1, Line 1-4) to the following:

“where J_1 , J_2 , and J_3 are the exchange coupling parameters; and similar notation applies to the DM vectors (D_1 , D_2 , and D_3), as shown in Fig. 1d. M denotes the magnetic moment of each atom, and K is the perpendicular magnetic anisotropy constant. The calculation method of the exchange coupling parameter, ...”

2. In LaCl/P- configuration, skyrmions can be created or annihilated by tuning the anisotropy energy K from -0.5 meV to -0.76 meV. Since the appearance of skyrmion is very sensitive to the calculated parameters, it is hard to believe that the control of skyrmion using ferroelectric polarization can be as good as claimed by the authors.

Reply: This comment is highly appreciated, and it actually helped us to spot a typo in the previous manuscript. We found that we have mistaken $K = -0.1$ meV for $K = -0.5$ meV. In fact, there was no skyrmion when $K = -0.5$ meV, as shown in Fig. R1. We emphasize that this typo does not affect our discussion and the conclusion; it is only to show that the bimerons cannot appear in P^- state because of the large K .

Emergence of magnetic skyrmion is the result of competition between various magnetic energies; skyrmions debut only when these energies are within a specific range. In LaCl/ P^- configuration, we calculated the K values on which bimeron can occur are ranging from -0.04 meV to -0.13 meV (other parameters remain unchanged). Such a specific range of K is required for skyrmions to appear. According to Fig. 4b of the manuscript, the enhanced P^- state polarization can significantly extend the value of K to go beyond the aforementioned critical range, leading to bimeron's annihilation in this polarized state. This is a strong indication that skyrmions can be created or annihilated by tuning the ferroelectric polarization.

In the revised manuscript, we have added the following discussion (page 7, paragraph 1, Line 4-10) to further elucidate this point

Fig. R1. The top views of the micromagnetic simulation for LaCl/P- HS with $K = -0.5$ meV.

“Subsequently, we found that bimeron can appear when the K value ranges from -0.04 to -0.13 meV (other parameters remain unchanged). Fig. 2c represents the bimeron texture when the $K = -0.1$ meV, showing a smaller bimeron size compared to the $P+$ state and this phenomenon is explained in part 5 of the Supplementary Information. In the $P-$ state, the enhanced K value caused by the polarization completely exceeds the range that allows bimeron to exist, leading to bimeron’s annihilation in this polarized state.”

3. Figures 2a and 2c show the simulation of skyrmions for LaCl/P+ and LaCl/P- with $K = -0.5$ meV. Why do the skyrmions for the two cases have different sizes?

Reply: The parameters in two different polarization states are given in Table 1. Fig. 2a is for the case of LaCl/P+, whereas Fig. 2c is for the case of LaCl/P- with $K = -0.1$ meV. These two different sets of parameters (J , D , K , and M_s) give rise to skyrmions of different sizes.

Here, we make it clear in the caption of Fig. 2c and add a new sentence to remove this ambiguity “The varied parameters including J , D , K and M_s for $P+$ and $P-$ states are listed in Table 1, respectively.”

According to an early work by Sampaio *et al* [Sampaio, J. *et al*, stability and current-induced motion of isolated magnetic skyrmions in nanostructures. *Nature nanotechnology* 8, 839-844 (2013)), a larger D lead to a larger skyrmion. We carried out a similar test on bimeron in $P+$ state: By decreasing the value of D , we found that skyrmions do shrink. To answer Referee’s question, we have added the following discussions in part 5 of the Supplementary Information.

“Fig. 2c represents the bimeron texture when the $K = -0.1$ meV, showing a smaller bimeron size compared to the $P+$ state and this phenomenon is explained in part 5 of the Supplementary Information.

Part 5. The Relationship between bimeron size and D

In the nanodisk, the size of Néel-type skyrmions will increase with the increase of the D value [3]. However, whether this conclusion applies to bimeron is unknown. Here, we plot the bimeron size as a function of D value. It can be clearly seen from Fig. S5 that the size of bimeron indeed increases with the increase of D value.”

Fig. S5. The bimeron size in relation to D in the relaxed sample, other parameters are setting the same as LaCl/ $P+$ scenario.

4. In the ferroelectric polarization dependent LaCl magnetic properties section, the authors simulated an enhanced polarization of 3P using three layers of In_2Se_3 . Since the multiferroic property combining FM and FE materials has a nature of proximity effect which is very sensitive on interlayer distance, is it justified to simply add up the ferroelectric polarization of three In_2Se_3 layers and consider it the polarization experienced by the LaCl layer. In a Zhao et al, Nature Materials, 19, 503–507(2020), the electric properties of WTe_2 layer is affected by tuning the magnetization of CrI_3 layer. From their results, the electric conductance of WTe_2 is majorly dependent on the nearest CrI_3 layer.

Reply: We thank Referee for sharing with us her/his insight and raising this very interesting point. According to the work by Ding et al [Ding, W. et al. Prediction of intrinsic two-dimensional ferroelectrics in In_2Se_3 and other III2-VI3 van der Waals materials. Nature communications 8, 1-8 (2017)], In_2Se_3 polarization increases as a function of the film thickness and saturates as the thickness increases to three layers. In other words, the polarization can be enhanced by increasing the layer thickness within three layers. On the other hand, in the report of Zhao et al, control of WTe_2 conductance by CrI_3 in $\text{WTe}_2/\text{CrI}_3$ HS is attributed to the backscattering mechanism in topological insulator, that is, WTe_2 's conductance is affected by the excitation of magnons within the nearest CrI_3 layer. The conductance of WTe_2 depends on the

nearest CrI₃ layer and is not affected by the CrI₃ thickness.

What happens in our system is different. The control of LaCl magnetism in our LaCl/In₂Se₃ system is caused by electric field effect, which is fundamentally different from that in WTe₂/CrI₃ HS. When In₂Se₃ is in contact with LaCl, charge redistributes on the LaCl side to screen the built-in electric field in In₂Se₃. The substrate with larger polarization will make the charge redistributions of LaCl move more obvious, thus leading to a more distinct magnetism change for LaCl. We found from Fig. 4 of the manuscript that the In₂Se₃ thickness indeed plays a role in shaping the magnetic properties of LaCl.

We have added in the revised manuscript a paragraph (page 8, paragraph 2, Line 1-9) to address the difference between our system and the one discussed in the aforementioned reference,

“In our system, the out-of-plane polarization of the In₂Se₃ layer redistributes charges in the LaCl layer to screen the polarization field, resulting in a change in the magnetic properties of LaCl. The enhanced polarization is thus expected to improve the regulation for LaCl magnetism. For the ferroelectric substrate In₂Se₃, polarization increases as a function of the film thickness and saturates as the thickness increases to three layers [26]. Therefore, using a three-layered In₂Se₃, we simulated an enhanced polarization which is denoted as $3P_{\pm}$ for simplicity. Note that the method by increasing the layer thickness for better effect does not apply to all 2D HS system. For example, in the WTe₂/CrI₃ HS, the conductance of WTe₂ depends on the nearest CrI₃ layer and is not affected by the CrI₃ thickness [63].”

63. Zhao, W. *et al.* Magnetic proximity and nonreciprocal current switching in a monolayer WTe₂ helical edge. *Nature Materials* **19**, 503-507 (2020).

5. In figure 4 a, b and e, it is shown that the exchange coupling constants, the magnetic anisotropy energy and magnetic moment are largely affected when the polarization is applied in the -z direction and relatively insensitive when the polarization is aligned in the +z direction. In contrast, the Curie temperature increases symmetrically with increasing polarization magnitude in both directions. Since Curie temperature depends on the magnetic properties such as exchange energy, why the magnetic properties (exchange coupling, magnetic anisotropy and magnetic moment) and Curie temperature behave differently when ferroelectric polarization is applied in different directions?

Reply: In our heterostructures, K and magnetic moment are indeed relatively

insensitive to the polarization when the polarization is applied in the $+z$ direction ($P+$ state). However, the exchange coupling constant is largely affected in two different polarized state, as shown in Fig. 4a. Because of the change in the exchange coupling constant over 1 meV, the T_c has changed significantly. Fig. 4a shows that, compared to the free-standing LaCl monolayer, in $P+$ state, J_1 is very insensitive to the polarization, yet both J_2 and J_3 are enhanced greatly, which contributes to the increase in T_c . In the $P-$ state, although J_1 is enhanced, the values of J_2 and J_3 , which are larger than J_1 , are reduced. Therefore, the scenario of $P+$ state is more conducive to the increase of T_c .

In addition, coordination number is also an important factor that needs to be considered. Here, the larger J value with large coordination number leads to higher T_c . For any La ion, it interacts with three out-of-plane La ions and six in-plane ones, as shown in Fig. R2. Therefore, the in-plane exchange coupling contributes more to the determination of T_c . In other words, T_c in the $P+$ state is expected to be higher than that in $P-$ state. To incorporate Referee's comment, we have added the following discussions in part 3 of the Supplementary Information.

Fig. R2. The coordination number of La ion, same plane La ions are represented by the same color.

“We are thus able to tune the magnetism of the HS from FM LaCl/(3) $P+$ to paramagnetic LaCl/(3) $P-$ by the polarization when the temperature is in the range between 92 K (102 K) and 99 K (110K). **The discuss the T_c in details in part 3 of the Supplementary Information.**

In our HSs, the T_c in (3) $P+$ state is higher than (3) $P-$ state. Compared to the free-standing LaCl monolayer, the increase in the T_c mainly originate from the distinct increase of J_2 and J_3 exchange coupling under the (3) $P+$ state, due to the K , J_1 , and magnetic moment are relatively insensitive to the polarization. In the (3) $P-$ state, although J_1 is also enhanced, the values of J_2 and J_3 , which are larger than J_1 , are reduced. Therefore, the scenario of (3) $P+$ state is more conducive to the increase of the T_c . In addition, coordination number is also an important factor that needs to be considered, here, the larger J value with large coordination number leads to higher T_c . For any one La ion, it will interact with three out-of-plane La ion and six in-plane La ion, as shown in Fig. S3. Combined with Fig.4a, the in-plane exchange coupling effect contributes more to T_c . In other words, the T_c in $P+$ state is higher than $P-$

state.”

6. There are many typos, grammatical errors and wrongly labelled figure indexes in the manuscript. For example, 1. on line 1, page 8, “to mobilize the bimeron, see Fig. 4.” Fig. 4 should be Fig. 2g; 2. Figure caption of fig 2f should be 2g; 3. On last line, page 9, “underling” should be underlying; etc. Please check the manuscript carefully and correct all the errors.

Reply: We truly appreciate Referee’s comment on the writing of the manuscript. The language and writing in the revised manuscript have been improved significantly. We sincerely hope that Referee will enjoy reading the revised version. We have made the following specific changes to address the red flags raised by Referee:

“An in-plane current of $j = 3 \times 10^{10}$ A m⁻² was later applied along the x direction to mobilize the bimeron, see Fig. 2g.”

Fig. 2. The top views of the micromagnetic simulation for (a) LaCl/P+ and (b) LaCl/P- HSs. (c) The top views of the micromagnetic simulation for LaCl/P- HS with $K = -0.1$ meV, the red rectangle shows two bimerons with opposite topological numbers (d-f) Blue-Red color represents the region of x, y, z component along the magnetic direction of a bimeron. (g) Snapshots of the bimeron moving under a current of $j = 3 \times 10^{10}$ A m⁻² on nanotracks.

We performed a layer-resolved partial density calculation on the LaCl/3P± configurations to scrutinize the underlying mechanism.”

Reply to the third referee’s report:

The authors reported their theoretical findings that skyrmions can be generated, manipulated, and destroyed in heterostructures consisting of a ferromagnetic monolayer LaCl and a ferroic layer In2Se3. They combined 1) DFT calculations, 2) OOMMF micromagnetic simulations, and 3) finite temperature Monte Carlo simulations to show that, by controlling the polarization state of the heterostructure, one can manipulate T_c , the type of anisotropy, and skyrmion populations of the heterostructure. Interestingly, skyrmions are found to exist in the form of bimerons when LaCl has easy-plane anisotropy and they can be driven by a current. Creation/annihilation/manipulation of skyrmions in 2D magnets is an important topic in spintronics and using the heterostructures of 2D magnets and 2D ferroics is a promising direction, and I believe that this work reports the first comprehensive theoretical prediction of a suitable material for that purpose. Therefore, I recommend the publication of the manuscript in Nature Communications. However, I have a couple of minor comments, which should be addressed before the publication.

Reply: We are genuinely delighted by Referee's recommendation for publication. Meanwhile, we are grateful to Referee for sharing with us her/his insight in this exciting field.

1) English usage is poor. Extensive language editing seems necessary.

Reply: We have made significant effort to edit the language and improve the writing in the revised manuscript. We hope that Referee find it enjoyable to read the new version.

2) Two important references are missing. [Park et al., "Néel-type skyrmions and their current-induced motion in van der Waals ferromagnet-based heterostructures," arXiv:1907.01425], where Neel type skyrmions are experimentally observed and driven in 2D magnet / h-BN structures, and [Ding et al., "Observation of Magnetic Skyrmion Bubbles in a van der Waals Ferromagnet Fe₃GeTe₂," Nano Lett. 20, 868 (2020)] where skyrmions are observed in a 2D ferromagnet.

Reply: To incorporate Referee's suggestion, we have included these two important references in our revised manuscript (page 2, paragraph 1, Line 6-9).

"In recent experiments skyrmion and its motion under electric current have been observed in Fe₃GeTe₂ [16,17], proving 2D magnetic materials a new category of skyrmion medium. In addition, another study showed that..."

16. Ding, B. et al. Observation of Magnetic Skyrmion Bubbles in a van der Waals ferromagnet Fe₃GeTe₂. Nano Letters 20, 868-873 (2019).

17. Park, T.-E. et al. Observation of magnetic skyrmion crystals in a van der Waals ferromagnet Fe₃GeTe₂. arXiv preprint arXiv:1907.01425 (2019).

3) About the current-induced skyrmion motion, it is not clear what type of spin torque is used. Is it spin-transfer torque or spin-orbit torque?

Reply: Indeed, this comment from Referee helps us to further elucidate this point. We have clarified in the revised manuscript (page 12, paragraph 2, Line 6-7) that the spin torque is Slonczewski-like spin-transfer torque.

“We include in the simulation a Slonczewski-like spin-transfer torque (STT) driven by the spin current that is generated by the spin-Hall effect, ...”

4) *The discussion of the topological charge of observed skyrmions is missing. For example, a) can skyrmions of both $Q = 1$ and $Q = -1$ be created? Or only one type of skyrmions can be created? Also, b) What effects does changing the polarization have on the topological charge of stable skyrmions?*

Reply: We thank Referee for these two questions which helped us reposition our arguments. Shorter answers to Referee’s questions are as follows:

a) Skyrmions of both $Q = 1$ and $Q = -1$ can be created, as shown in Fig. 3c.

b) The topological charge number Q changes from 1 to 0 when switching the polarized state.

Meanwhile, we have added in the revised manuscript (page 7, paragraph 1) the discussion on bimeron topological charge.

“We used OOMMF software to perform micromagnetic simulations of the LaCl monolayer and LaCl/ P_{\pm} HSs in a nanodisk geometry with a diameter of 200 nm. The varied parameters including J , D , K and M_s for P_+ and P_- states are listed in Table 1, respectively. The initial spin state was set to paramagnetic (random) states. In the LaCl/ P_+ configuration, bimerons spin texture emerges, as shown in Fig. 2a. The calculation shows the topological charge of the bimeron $Q = 1.0$, consisting of a vortex ($Q = 0.5$) and an anti-vortex ($Q = 0.5$). The extracted bimeron texture map is depicted in Fig. 2d-e. Due to the presence of in-plane anisotropy, bimeron’s outer magnetization is along the in-plane direction rather than the out-of-plane one, which consequently allows two bimerons with opposite topological numbers ($Q = 1$ and $Q = -1$) to coexist in the same magnetic domain (see Fig. S4) [61]. This is significantly different from other types of skyrmions. This spin texture disappears, accompanied by a vanishing Q to zero when the polarization is reversed to the P_- state, shown in Fig. 2b. Subsequently, we found that bimeron can appear when the K value ranges from -0.04 to -0.13 meV (other parameters remain unchanged). Fig. 2c represents the bimeron texture when the $K = -0.1$ meV, showing a smaller bimeron size compared to the P_+ state and this phenomenon is explained in part 5 of the Supplementary Information. In the P_- state, the enhanced K value caused by the polarization completely exceeds the range that allows bimeron to exist, leading to bimeron’s annihilation in this polarized state. Furthermore, Fig. 3c verifies the above conclusion that bimerons can appear in the same magnetic domain. Therefore, by simply switching the FE polarization, two distinct magnetic states - referred to in the binary code as “0” and “1” - can be realized and could be used as building blocks for information storage. The diameter of bimeron in LaCl/ P_+ HS is only about 23 nm,

which enhances the controllability and integratability of the bimerons-based functional devices.”

Reviewers' Comments:

Reviewer #1:

Remarks to the Author:

I am satisfied with author's reply.

Reviewer #2:

Remarks to the Author:

I have reviewed the comments and changes to the manuscript, the authors have addressed my concerns in the last review. I therefore recommend its publication.

Reviewer #3:

Remarks to the Author:

The authors responded to my comments and, I believe, to the other referees' comments as well. I thus recommend the publication of the manuscript in Nature Communications.

Dear Editor,

Thank you very much for your valuable time and great effort. We are delighted to hear of the acceptance of our manuscript NCOMMS-20-20354B. We would like to express our sincere appreciations to you and the referees for positive evaluations, efficient work and kindly help during the review process of manuscript.

We have revised the manuscript and believed we may have addressed all the suggestions in the checklist and thus the revised manuscript in its present form is ready for publication in Nature Communication.

Sincerely yours,
Zhenxiang Cheng

REVIEWERS' COMMENTS

Reviewer #1 (Remarks to the Author): I am satisfied with author's reply.

Reply: We thank the reviewer for his/her insightful comments and suggestions during the review process. We would like express our gratitude for the referee recommending our manuscript for publication.

Reviewer #2 (Remarks to the Author): I have reviewed the comments and changes to the manuscript, the authors have addressed my concerns in the last review. I therefore recommend its publication.

Reply: We thank the referee for spending valuable time to carefully read the manuscript. We also thank him/her for constructive, professional comments and suggestions during the review process. His/her comments are indeed very helpful to improve the quality of the manuscript.

Reviewer #3 (Remarks to the Author): The authors responded to my comments and, I believe, to the other referees' comments as well. I thus recommend the publication of the manuscript in Nature Communications.

Reply: We appreciate the referee for his/her timely response and positive evaluation.

We thank the referee valuable time and professional suggestions to enhance the quality of manuscript.